# Preparation and Properties of Thin-Film Composite Forward Osmosis Membranes Supported by Cellulose Triacetate Porous Substrate via a Nonsolvent-Thermally Induced Phase Separation Process

**DOI:** 10.3390/membranes12040412

**Published:** 2022-04-10

**Authors:** Jian-Chen Han, Xiao-Yan Xing, Jiang Wang, Qing-Yun Wu

**Affiliations:** 1School of Chemical Engineering and Technology, Sun Yat-sen University, Zhuhai 519082, China; hanjch5@mail2.sysu.edu.cn; 2Faculty of Materials Science and Chemical Engineering, Ningbo University, Ningbo 315211, China; xxy18755023221@126.com (X.-Y.X.); 15757460305@163.com (J.W.)

**Keywords:** thin-film composite membrane, forward osmosis, phase inversion, porous substrate, internal concentration polarization

## Abstract

A porous substrate plays an important role in constructing a thin-film composite forward osmosis (TFC-FO) membrane. To date, the morphology and performance of TFC-FO membranes are greatly limited by porous substrates, which are commonly fabricated by non-solvent induced phase separation (NIPS) or thermally induced phase separation (TIPS) processes. Herein, a novel TFC-FO membrane has been successfully fabricated by using cellulose triacetate (CTA) porous substrates, which are prepared using a nonsolvent-thermally induced phase separation (N-TIPS) process. The pore structure, permeability, and mechanical properties of CTA porous substrate are carefully investigated via N-TIPS process (CTA_N-TIPS_). As compared with those via NIPS and TIPS processes, the CTA_N-TIPS_ substrate shows a smooth surface and a cross section combining interconnected pores and finger-like macropores, resulting in the largest water flux and best mechanical property. After interfacial polymerization, the obtained TFC-FO membranes are characterized in terms of their morphology and intrinsic transport properties. It is found that the TFC-FO membrane supported by CTA_N-TIPS_ substrate presents a thin polyamide film full of nodular and worm-like structure, which endows the FO membrane with high water permeability and selectivity. Moreover, the TFC-FO membrane supported by CTA_N-TIPS_ substrate displays a low internal concentration polarization effect. This work proposes a new insight into preparing TFC-FO membrane with good overall performance.

## 1. Introduction

Membrane-based technology is booming in order to deal with the worldwide water scarcity due to its superior and distinct advantages of high water quality, excellent separation efficiency, and environmental friendliness [1,2,3,4]. Among others, forward osmosis (FO) refers to a promising membrane-based process for water purification and seawater desalination [5,6,7]. The FO process is driven by osmotic pressure difference, during which water molecules spontaneously pass through a permselective membrane from a feed solution to a draw solution. Thus, the FO process is characterized by low energy consumption and a low membrane-fouling tendency as compared with external pressure-driven membrane-based processes [8,9].

Nowadays, thin-film composite (TFC) membrane is the most commonly used FO membrane, consisting of a selective layer and a porous substrate [10,11,12]. Among others, the porous substrate plays an important role in regulating the structure and the performance of TFC-FO membrane. For example, the surface-pore size and porosity of the substrate deeply affects the morphology of the selective layer through interfacial polymerization [13]. On the other hand, the porous substrate as the mass transfer channels is directly connected to the water permeability. An internal concentration polarization (ICP) will be caused by the diluted or concentrated solution inside the porous s ubstrate, which brings about the decreasing of the effective osmotic pressure difference between the two sides of the selective layer [14,15,16,17,18]. As a consequence, the water flux of TFC-FO membrane is severely reduced. ICP is mainly dependent on the substrate structure, which is described by the structural parameter (*S*). ICP can be alleviated by using a substrate with small thickness (*l*), high porosity (*ε*), and low tortuosity (*τ*) (*S = l*·*τ/ε*). Therefore, a satisfactory porous substrate is key for a TFC-FO membrane with excellent comprehensive performance, such as high water permeability, low reverse salt flux, high selectivity, and good physical strength.

By far, a porous substrate is mainly constructed through a non-solvent induced-phase separation (NIPS) process, which is carried out at low temperature and has high accessibility in manufacturing [19,20]. The substrates prepared using a NIPS process usually have a smooth surface with nanoscale pores suitable to be composite with the PA layer through interfacial polymerization. However, an asymmetric structure as the characteristic of the substrate via NIPS always results in a poor mechanical strength. In contrast, the substrates via thermally induced phase separation (TIPS) method show a symmetric structure and a good physical strength [21,22]. Yabuno et al., prepared a TFC-FO membrane using polyvinylidene difluoride substrate via TIPS process, which presented a higher physical strength compared to that using polysulfone substrate via NIPS method [21]. Recently, nonsolvent-thermally induced phase separation (N-TIPS) has been put forward to tailor the substrate structure by combining both advantages of NIPS and TIPS processes [23,24,25,26]. N-TIPS process not only can be operated at moderate condition, but also can be modulated by various parameters. Moreover, the substrate via N-TIPS demonstrates a smooth and porous surface, symmetric structure, and good mechanical strength, which are beneficial for TFC-FO membrane with excellent performance. However, using a N-TIPS method to prepare TFC-FO membrane substrate still lacks deep exploration. It is worthwhile to develop a new kind of TFC-FO membrane supported by the porous substrates via N-TIPS, and further investigate its structure–performance relationship.

Herein, a novel TFC-FO membrane was fabricated with a cellulose triacetate (CTA) porous substrate via N-TIPS method. CTA is a popular membrane material derived from the natural cellulose. Thus, CTA membranes attract extensive attention due to their great biocompatibility, excellent hydrophilicity, and outstanding antifouling property compared with most other polymeric membranes [27,28,29,30]. In this work, we comprehensively investigated the structure and property of CTA substrate via N-TIPS, and compared with those via NIPS and TIPS processes. Moreover, the morphology and performance of the TFC-FO membranes were also studied in detail and correlated with the corresponding porous substrates. Detailed discussion was focused on the water permeability, reverse salt flux, and structure parameter of TFC-FO membranes in light of the porous substrates prepared by different phase separation processes. The obtained TFC-FO membrane with CTA substrate via the N-TIPS process demonstrated excellent comprehensive performance that could be promising for many applications.

## 2. Experimental Section

### 2.1. Materials

Cellulose triacetate (CTA, *M_w_* = 1.2 × 10^5^ Da; Acros Organics, Shanghai, China) was dried under vacuum at 60 °C for 4 h to remove moisture before usage. Dimethyl sulfone (DMSO2, 99% purity) was utilized as a solvent, supplied by Dakang Chemicals Co., Hangzhou, China. Polyethylene glycol (PEG400, *M_w_* = 380~430, AR grade) as an additive was employed by Sinopharm Chemical Reagent Co. Ltd., Shanghai, China. Polyvinyl alcohol (PVA, *M_w_* = 78,243~78,475) and trimesoyl chloride (TMC, ~98%) were supplied by Aladdin. *m*-Phenylenediamine (MPD, >99%, Acros Organics, Shanghai, China) and Isopar-G (Exxon Mobil Chemical Co., Shanghai, China) were used as received. Other chemicals including sodium hypochlorite (NaClO), sodium bisulfate (NaHSO_3_) and sodium chloride (NaCl) were purchased from Sinopharm Chemical Reagent Co. Ltd., Shanghai, China.

### 2.2. Fabrication of TFC-FO Membranes

#### 2.2.1. Preparation of CTA Porous Substrates

First, 10 wt% CTA was added into DMSO2/PEG400 mixtures (the weight ratio was 70:30) and stirred at 160 °C to obtain a homogeneous solution. After it was degassed, the casting solution was rapidly poured onto a polyester-screen-wrapped mold, which was preheated at 140 °C. Then, a liquid film was cast on the polyester screen using a casting knife with a height of 200 μm. It should be mentioned that the polyester screen was pretreated by infiltration in a 5 wt% PVA solution for 30 min. Then, three kinds of phase-separation methods were introduced to obtain different CTA porous substrates, and the corresponding preparation conditions were listed in Table 1 according to our previous work [22,30,31]. CTA_TIPS_, CTA_N-TIPS_, and CTA_N-TIPS_ represent the CTA porous substrates prepared via TIPS, NIPS, and N-TIPS methods, respectively.

#### 2.2.2. Interfacial Polymerization on Porous Substrates

The polyamide (PA) selective layer was synthesized on the CTA porous substrate by interfacial polymerization (IP) of MPD and TMC. First, the 3.4 wt% MPD aqueous solution was poured on the CTA porous substrate for 2 min, and then the excess solution was removed. Second, the 0.15 wt% TMC solution in Isopar-G was poured onto the porous substrate for 1 min to induce IP reaction. Third, the membrane was dried in air for 2 min after removing the excess organic solution, and subsequently treated at 90 °C for 4 min. Fourth, the membrane was successively dipped into NaClO aqueous solution and NaHSO_3_ aqueous solution for 2 min. At last, the TFC-FO membrane was finalized by heated at 90 °C for 4 min. The TFC-FO membranes were thoroughly rinsed and stored in DI water at 4 °C before test.

### 2.3. Membrane Characterization

The surface and cross-section morphologies of membranes were observed by a field emission scanning electron microscope (FESEM, Sirion-100, FEI, Eindhoven, The Netherlands) after sputtered with a thin layer of platinum (JFC-1100, JEOL, Tokyo, Japan). The surface pore size was determined by analyzing the FESEM images. The surface morphology and roughness of membranes were investigated by atomic force microscope (AFM, Dimension 3100V, Veeco, New York, NY, USA) with a scanning area of 5 μm × 5 µm. A Fourier transform infrared spectroscope in the attenuated total reflectance mode (ATR-FTIR, Nicolet 6700, Thermo Fisher Scientific, Waltham, MA, USA) was applied to identify the surface chemical compositions of membranes. The mechanical properties of the porous substrates were measured using a universal testing machine (Instron 5566 instrument, Norwood, MA, USA) at room temperature. Rectangular samples (1 cm × 3 cm) were extended at a constant elongation rate of 2 mm/min to determine the tensile stress and elongation at breaking point. All the data were the average of five repeat tests in order to reduce the error.

Pure water fluxes (*J_w,substrate_*) of CTA porous substrates were examined by a pressure-driven dead-end filtration device (XX677P05, Millipore, Burlington, MA, USA) at room temperature. The porous substrate was prepressed at 0.12 MPa for 30 min and then stabilized for 5 min at 0.10 MPa before test. Each datum was the average of at least three parallel experiments. The *J_w,substrate_* values were calculated according to the following Equation (1).
(1)$$J_{w,\;substrate}=\frac{V}{A\times \Delta t}\times 100\%$$
where *V* is the volume of permeated water (L), *A* is the effective membrane area (m^2^), and Δ*t* is the permeation time (h).

The porosity (*P*) of the CTA porous substrate was evaluated by gravimetric method according to our previous work, which was calculated from Equation (2) [32].
(2)$$P=\frac{\left(w_{0}-w_{1}\right)/\rho_{\mathrm{water}}}{\left(w_{0}-w_{1}\right)/\rho_{\mathrm{water}}+w_{1}/\rho_{p}}\times 100\%$$
where *ρ_p_* and *ρ*_water_ are the density of CTA (1.3 g/cm^3^) and water at 25 °C (1.0 g/cm^3^), respectively. *w*_0_ and *w*_1_ are the weights of substrate (g) before and after absorbing water, respectively. Each sample was measured three times and the final result was the average value.

### 2.4. FO Performance Test of TFC Membrane

The water flux (*J_w_*, L/m^2^·h, abbreviated as LMH) and reverse salt flux (*J_s_*, g/m^2^·h, abbreviated as gMH) of TFC-FO membranes were tested using a lab-scale cross-flow FO system with a flow rate of 2.0 L/h. A 1.0 M NaCl solution was used as the draw solution while DI water was used as the feed solution at 25 ± 1 °C. An effective membrane area of 4.19 cm^2^ was applied during each test for both AL-FS and AL-DS membrane orientations.

The water flux (*J_w_*) and the reverse salt flux (*J_s_*) can be calculated by Equations (3) and (4).
(3)$$J_{w}=\frac{\Delta m/\rho }{A\cdot \Delta t}$$
(4)$$J_{S}=\frac{\Delta (C_{t}\times V_{t})}{A\cdot \Delta t}$$
where Δ*m* is the weight change of feed solution, which was monitored by a computer connected to a balance; *A* is the effective membrane surface area; *ρ* is the density of feed solution; Δ*t* is the measuring time; and *C_t_* and *V_t_* are the salt concentration and the feed volume at the end of the predetermined experiment duration, respectively.

The water permeability coefficient (*A*), salt rejection (*R_s_*), and salt permeability coefficient (*B*) were measured by using a cross-flow reverse-osmosis filtration setup. The TFC-FO membrane was fixed in a stainless-steel filtration cell and prepressed for 30 min under a pressure of 6 bar before test. The effective membrane area (*S_m_*) was 4.91 cm^2^ and the cross-flow velocity was fixed at 3.0 L/min. All tests were conducted with 200 ppm of NaCl solution at a pressure (Δ*P*) of 5 bar. The values of *A*, *R**_s_* and *B* were calculated from Equations (5)–(7), respectively.
(5)$$A=\frac{\Delta V}{\Delta t\cdot S_{m}\cdot \Delta P}$$
(6)$$R_{S}=\frac{C_{f}-C_{p}}{C_{f}}\times 100\%$$
(7)$$B=A\times \frac{\left(1-R_{S}\right)\cdot \left(\Delta P-\Delta \pi \right)}{R_{S}}$$
where Δ*V* and Δ*t* are the volume of permeation water and the operating time, respectively. *C_f_* and *C_p_* represent the salt concentrations in the feed and permeate solution, which were determined by conductivity measurement. Δ*π* is the osmotic pressure difference across the membrane under RO mode.

Then, the structural parameter (*S*) can be calculated from Equation (8), where *J_W_*, *D*, *π_D,b_*, and *π_F,m_* are the water flux measured under AL-FS, the bulk diffusion coefficient of NaCl in aqueous solution, the bulk osmotic pressure of the draw solution, and the osmotic pressure at the membrane surface on the feed solution side, respectively.
(8)$$S=\frac{D}{J_{W}}\ln(\frac{B+A\times \pi_{D,b}}{B+Jw+A\times \pi_{F,m}})$$

## 3. Results and Discussion

### 3.1. Structure and Property of CTA Porous Substrates

Figure 1 shows the structure of CTA porous substrates prepared by TIPS, NITS, and N-TIPS methods. The CTA_TIPS_ porous substrate obviously shows a symmetric cross-section full of sponge-like pores, which is the typical morphology of membrane prepared by TIPS process. Herein, the CTA_TIPS_ substrate was prepared by immersing the CTA/DMSO2/PEG400 solution at 160 °C into a cooling bath of glycerin at 50 °C. It should be mentioned that as the solvent of CTA, DMSO2 is immiscible with glycerin, and thus the mass exchange between DMSO2 and glycerol could never happen. On the other hand, the intense heat exchange promotes the crystallization of CTA and the solid-liquid phase separation, resulting in the granular structure in the cross-section and membrane surfaces.

The NIPS process can be induced by using DI water as the cooling bath, which is miscible with DMSO2. As the CTA/DMSO2/PEG400 solution at 160 °C was immersed into DI water, the mass exchange between DMSO2 and DI water immediately took place at the surface of the liquid membrane, and a dense surface layer formed. The intruded DI water further induced the phase separation across the whole CTA/DMSO2/PEG400 solution. Finally, finger-like macropores formed inside the membrane as the typical morphology of membranes prepared by NIPS process, which were found in the cross-section of CTA_NIPS_ and CTA_N-TIPS_ substrates (Figure 1a). The CTA_NIPS_ substrate completely eliminated the granular structure, while the CTA_N-TIPS_ substrate still retained a part of it (Figure 1b,d). This result can be ascribed to the temperature difference between the solution and the cooling bath that determines the competitive relationship between TIPS and NIPS processes. Specifically, the CTA_NIPS_ substrate was prepared in the cooling bath at 95 °C, in which the driven force for the crystallization of CTA was dramatically weakened, and thus the TIPS process was inhibited. As the temperature of the DI water decreased to 50 °C, both TIPS and NIPS processes were induced and the CTA_N-TIPS_ substrate was obtained. It can be seen that the sponge-like sublayer is thicker and the figure-like macropores are reduced in the CTA_N-TIPS_ substrate compared to the CTA_NIPS_ substrate.

The surface structure of the porous substrate plays an essential role in the interfacial polymerization and the polyamide morphology. As shown in Figure 1c, the CTA_TIPS_ substrate has a granular top surface due to the crystallization of CTA on the surface. As the NIPS process was introduced, the granule size decreases and the top surfaces of the CTA_NIPS_ and CTA_N-TIPS_ substrates become smooth. These results are further confirmed by AFM. Figure 2 shows that the CTA_N-TIPS_ substrate has the smoothest top surface with an average roughness value as low as 4.46 nm, much lower than those of CTA_TIPS_ (17.75 nm) and CTA_NIPS_ (9.24 nm).

The surface pore size and the surface porosity of the CTA porous substrates were counted and shown in Figure 3a. It is clear that the CTA_N-TIPS_ substrate had a surface pore size of 11.1 nm, similar to that of the CTA_TIPS_ substrate (11.7 nm). However, the surface porosity of CTA_N-TIPS_ substrate dramatically increased, which means that the NIPS process is beneficial to the surface pore formation through the mass exchange of DMSO2 and DI water. In contrast, the CTA_NIPS_ substrate had the largest surface pore size (18.8 nm) and highest surface porosity (5.4%). On the other hand, the overall porosity of the CTA_N-TIPS_ substrate was 71.9%, larger than that of the CTA_TIPS_ substrate (60.9%) and the CTA_NIPS_ substrate (67.7%) (Figure 3b).

As indicated in Figure 3b,c, the properties of CTA porous substrates were tested in terms of pure water flux, tensile strength, and elongation. The CTA_N-TIPS_ substrate displayed the largest water flux of 1124.0 L/m^2^·h, which was almost two times as large as that of CTA_NIPS_ substrate (602.4) and forty times larger than that of the CTA_TIPS_ substrate (29.7). The excellent water permeability of CTA_N-TIPS_ substrate is mainly contributed to its large overall porosity, porous top surface, and sponge-like pore structure. Figure 3c demonstrates the mechanical properties of CTA porous substrates. It is widely believed that high mechanical strength is one of the advantages of polymer membranes prepared using the TIPS method. Herein, the CTA_TIPS_ substrate showed higher tensile strength (17.4 MPa) than that of the CTA_NIPS_ substrate (9.8 MPa) due to its uniform pore structure and relatively low porosity. However, the elongation of CTA_TIPS_ substrate (13.5%) was a little lower than that of the CTA_NIPS_ substrate (14.6%), which is because the stacked granular structure may become the breaking point. In contrast, the CTA_N-TIPS_ substrate demonstrated the best tensile strength (18.2 MPa) and elongation (21.1%), which was beneficial for both the TIPS and the NIPS process. 

### 3.2. Surface Chemical Composition and Morphology of TFC-FO Membranes

ATR-FTIR was used to characterize the surface chemical composition of TFC-FO membrane and confirm the formation of a polyamide (PA) layer on the CTA substrate. As shown in Figure 4a, three absorption peaks at 1663 cm^−1^, 1610 cm^−1^, and 1542 cm^−1^ were detected, which represent the stretching vibration of -C=O (amide I), the aromatic amide bond, and the stretching band of C-N (amide II), respectively [22]. Moreover, the XPS spectra also indicated the formation of PA layer based on the signal of nitrogen element (Figure 4b). These results reveal that the IP reaction was conducted and a PA-selective layer was formed on all kinds of CTA porous substrates. Furthermore, the surface element content of TFC-FO membranes was further calculated by the XPS result (Table 2). According to the O/N ratio, the cross-linking degree of PA on TFC_NIPS_ membrane is higher than those of TFC_TIPS_ and TFC_N-TIPS_ membranes.

As shown in Figure 5a, all the TFC-FO membranes present a “ridge-and-valley” top surface as a typical morphology of the PA layer from interfacial polymerization. Moreover, a large leaf-like structure can be observed on TFC_NIPS_ membrane, whereas a nodular and worm-like structure is highly obvious on TFC_N-TIPS_ membrane. This result means more MPD exists on the CTA_N-TIPS_ and CTA_NIPS_ substrates than on the CTA_TIPS_ substrate, which is due to the relatively large surface pore size and surface porosity of the former (Figure 6). The adsorbed MPD within the porous substrates further diffuses out, reacts with the residual acyl chlorides, and forms a thick PA film [33]. Correspondingly, the PA layer of TFC_NIPS_ membrane has an average thickness of 300 nm, larger than that of TFC_N-TIPS_ (235 nm) and TFC_TIPS_ membranes (207 nm). The enlarged view of the cross-section reveals that the PA layers on the loose CTA_N-TIPS_ and CTA_NIPS_ substrates have intensively nodular protuberance, and voids existed inside the whole ridge-and-valley structure (Figure 5b). In contrast to the dense PA layer of TFC_TIPS_ membrane, the nodular protuberance and voids facilitate to increase the water transporting area and then enhance the water flux of TFC-FO membranes [34,35].

Moreover, AFM was also conducted to detect the surface morphology and the mean roughness (*R_a_*) of the TFC-FO membranes. As can be seen in Figure 7, the rutted and uneven surface is obvious on TFC_TIPS_ membrane, whereas the small bulges homogenously lay on TFC_NIPS_ membrane. This result is similar to that observed under SEM, as mentioned above. Correspondingly, the *R_a_* values for TFC_TIPS_ (198 nm) are much larger than those for TFC_NIPS_ (127 nm), and TFC_N-TIPS_ (110 nm). This result suggests that CTA porous substrates with a smooth surface and high surface porosity may lead to relatively smooth PA films.

### 3.3. FO Performance of TFC-FO Membranes

The FO performance of the fabricated TFC-FO membranes was evaluated in both AL-FS and AL-DS by using DI water as a feed solution and 1 M NaCl as a draw solution. As shown in Figure 8a, all the TFC-FO membranes displayed higher water fluxes in AL-DS than those in AL-FS due to the severe ICP effect in the AL-FS. Among others, TFC_N-TIPS_ (16.84 LMH in AL-DS and 14.89 LMH in AL-FS) and TFC_NIPS_ membranes (19.54 LMH in AL-DS and 16.94 LMH in AL-FS) presented relatively higher water fluxes than TFC_TIPS_ membrane (10.08 LMH in AL-DS and 8.41 LMH in AL-FS). Moreover, the reverse salt flux of TFC_N-TIPS_ membrane was much lower (4.67 gMH in AL-FS, and 10.03 gMH in AL-DS) than in TFC_N-TIPS_ and TFC_TIPS_ membranes (Figure 8b). Therefore, TFC_N-TIPS_ membrane demonstrates the smallest specific salt flux (*J_S_*/*J_W_* ratio), indicating the best selectivity for water molecules (Figure 8c).

The performance of TFC-FO membranes can be connected to the morphology and the surface composition of the PA layer, including the surface structure, the thickness, and the cross-linking degree [36,37]. Compared with TFC_TIPS_ membrane, the higher water flux for TFC_N-TIPS_ and TFC_NIPS_ membranes can be attributed to their intensive water-transporting area from the nodular protuberance and voids inside the PA layer, although their PA layers are thicker (Figure 5). On the other hand, the severe reverse salt flux of TFC_TIPS_ membrane may be ascribed to the low cross-linking degree of the PA layer. Additionally, the water flux of TFC-FO membrane is connected to the ICP effect of CTA porous substrate [11]. Table 3 lists the intrinsic transport parameters of TFC-FO membrane, including the pure water permeability coefficient (*A*), salt permeability coefficient (*B*), salt rejection (*R_s_*), and structural parameters (*S*). The TFC_N-TIPS_ and TFC_NIPS_ membranes have much smaller *S* values than TFC_TIPS_ membrane, which indicates a relatively low ICP effect in the former. This result can be contributed to the high porosity and low pore tortuosity of the CTA_N-TIPS_ and CTA_NIPS_ substrates, which can be indicated by the small *τ* values. The pure water permeability of the FO membrane was measured in a RO mode at 5 bar, using the DI water as the recycle solution. The *A* values of TFC_NIPS_ (1.03 L/m^2^·h·bar) and TFC_N-TIPS_ (0.90 L/m^2^·h·bar) membranes were 50~60% higher than that of TFC_TIPS_ (0.61 L/m^2^·h·bar) membrane. Meanwhile, the TFC_N-TIPS_ membrane demonstrated the highest NaCl rejection (92.6%) and smallest *B*/*A* value (0.39 bar), which reveals that the TFC_N-TIPS_ membrane has excellent selectivity. In summary, the TFC_N-TIPS_ membrane showed the best overall performance compared with the TFC_TIPS_ and TFC_NIPS_ membrane.

## 4. Conclusions

A novel thin-film composite forward osmosis (TFC-FO) membrane was fabricated using a cellulose triacetate (CTA) porous substrate prepared via a nonsolvent, thermally induced phase separation (N-TIPS) process. It was found that the CTA_N-TIPS_ substrate has a smooth surface and a cross-section combining interconnected pores and finger-like macropores, which endow the substrate with high water flux and good mechanical properties. Moreover, the corresponding TFC-FO membrane supported by CTA_N-TIPS_ substrate presented the best overall performance compared with those membranes supported by CTA substrates via a non-solvent induced-phase separation or thermally induced phase separation process. The TFC-FO_N-TIPS_ membrane not only displayed both high water permeability and selectivity, but also showed a low internal concentration polarization effect.

## Figures and Tables

**Figure 1 membranes-12-00412-f001:**
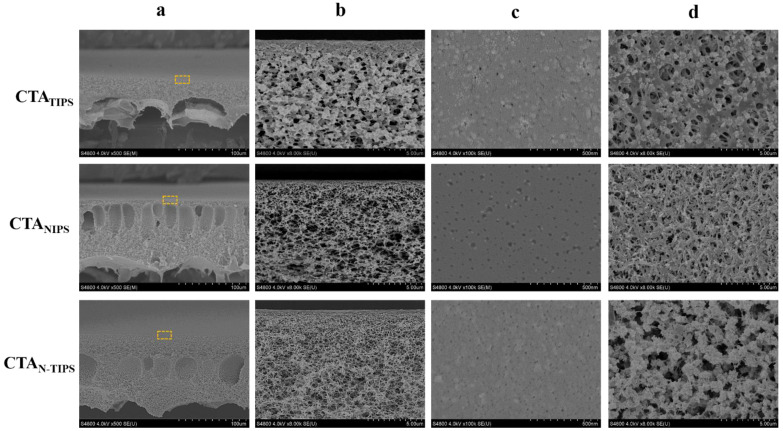
FESEM images of the cross-section (**a**,**b**); the top surface (**c**); and the bottom surface (**d**) of CTA porous substrates prepared by TIPS, NIPS and N-TIPS methods. The images of line b show the magnified morphology of the regions indicated by the rectangles in the images of line a.

**Figure 2 membranes-12-00412-f002:**
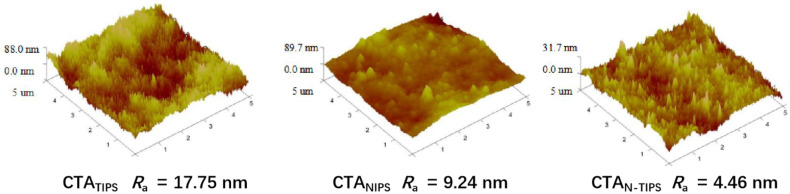
AFM images of CTA porous substrates.

**Figure 3 membranes-12-00412-f003:**
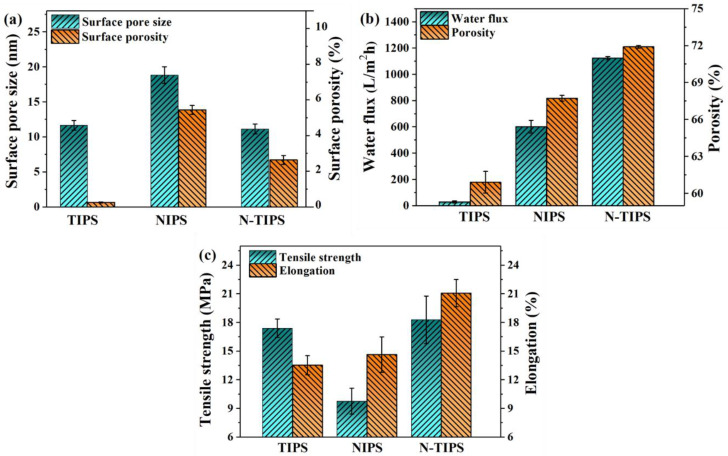
(**a**) Surface pore size and surface porosity of CTA porous substrates; (**b**) water flux and overall porosity of CTA porous substrates; (**c**) tensile strength and elongation of CTA porous substrates.

**Figure 4 membranes-12-00412-f004:**
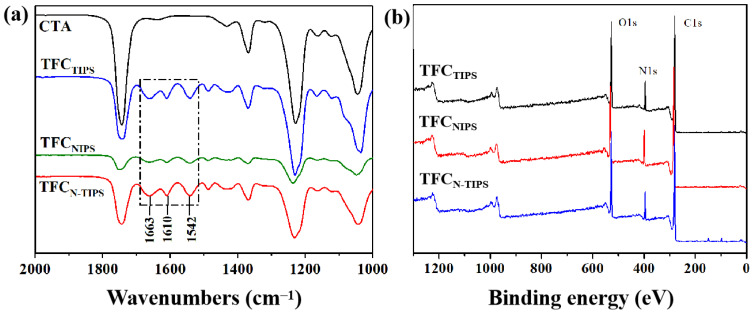
(**a**) ATR-FTIR spectra and (**b**) XPS spectra of TFC-FO membranes.

**Figure 5 membranes-12-00412-f005:**
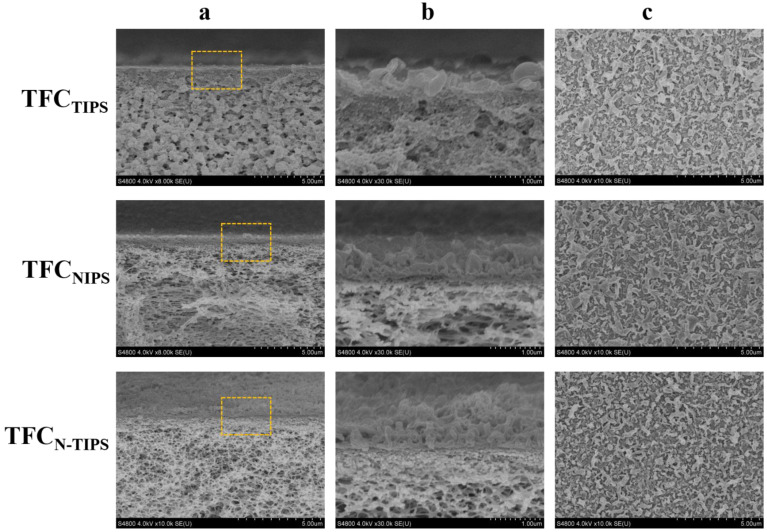
SEM images illustrating the cross-section (**a**,**b**) and top surface (**c**) of TFC-FO membranes. The images of line b show the magnified morphology of the regions indicated by the rectangles in the images of line a.

**Figure 6 membranes-12-00412-f006:**
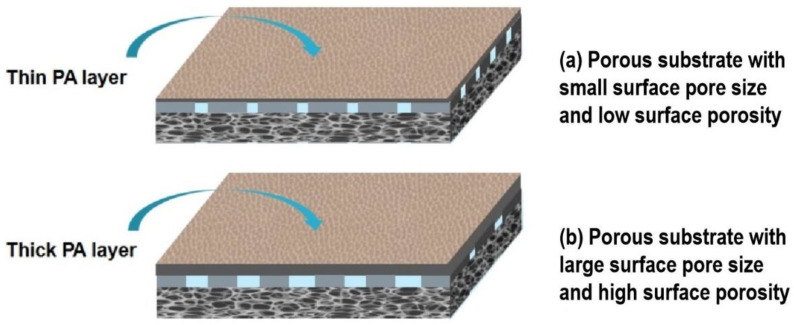
Schematic of PA layer formed on different porous substrates. (**a**) A substrate with small surface pore size and low surface porosity prefers to form a thin PA layer; (**b**) a substrate with large surface pore size and high porosity prefers to form a thick PA layer.

**Figure 7 membranes-12-00412-f007:**
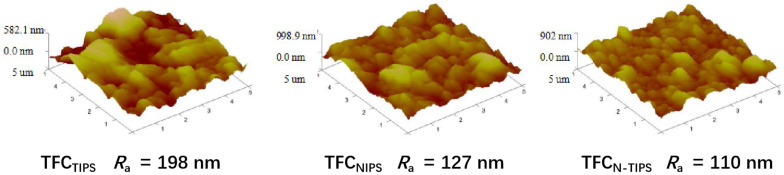
AFM images of TFC-FO membranes with different CTA porous substrates.

**Figure 8 membranes-12-00412-f008:**
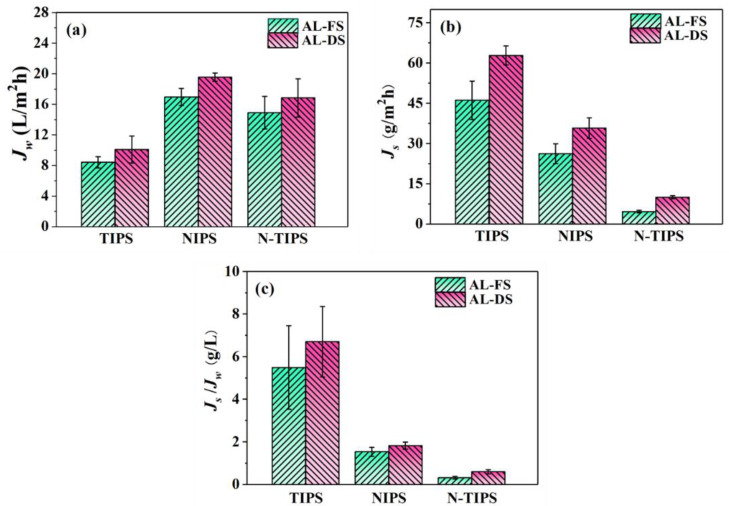
(**a**) Water flux; (**b**) reverse salt flux; and (**c**) specific salt flux of TFC-FO membranes at AL-DS and AL-FS modes. The draw solution and feed solution are 1.0 M NaCl and DI water, respectively. The cross-flow velocity is 20 L/h.

**Table 1 membranes-12-00412-t001:** Preparation conditions of different substrate membranes.

Membrane	Phase Separation Method	Coagulation Bath Composition	Coagulation Bath Temperature (°C)
CTA_TIPS_	TIPS	Glycerin	50
CTA_NIPS_	NIPS	DI water	95
CTA_N-TIPS_	N-TIPS	DI water	50

**Table 2 membranes-12-00412-t002:** Surface element content of TFC-FO membranes measured by XPS.

Samples	C (%)	N (%)	O (%)	S (%)	O/N
TFC_TIPS_	71.00	9.78	19.03	0.19	1.94
TFC_NIPS_	71.21	11.54	17.10	0.15	1.48
TFC_N-TIPS_	70.12	10.24	19.37	0.27	1.89

**Table 3 membranes-12-00412-t003:** The intrinsic transport parameters of TFC-FO membranes.

FO Membrane	*J_W_* (LMH)	*A* (L/m^2^·h·bar)	*R_S_* (%)	*B* (L/m^2^·h)	*B/A* (bar)	*τ/ε*	*τ*	*S* (μm)
TFC_TIPS_	8.41	0.61	66.21	1.53	2.51	4.27	320.8	737.3
TFC_NIPS_	16.94	1.03	84.84	0.91	0.88	2.41	163.2	337.7
TFC_N-TIPS_	14.89	0.90	92.63	0.35	0.39	2.75	197.6	384.8

## Data Availability

Not applicable.

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
