# Peer review of "Preparation and Properties of Thin-Film Composite Forward Osmosis Membranes Supported by Cellulose Triacetate Porous Substrate via a Nonsolvent-Thermally Induced Phase Separation Process"

_membranes, 2022, doi:10.3390/membranes12040412_

Round 1
Reviewer 1 Report
- In introduction, please expound why CTA was chosen as support membrane to investigate its membrane formation.
- From Figure 1, membrane prepared by NIPS looks rougher than membrane prepared by TIPS. Please recheck the AFM analysis.
- How did the author measure the membrane pore size? wet membrane has different pore size with dry membrane.
- In fact, high surface porosity membrane should have higher surface roughness. This data was not fit with AFM. Please check.
- Figure 4b, please translate to table for, and calculate the crosslinking degree. This work used MPD and TMC as monomer, the crosslinking degree can be easier to calculate from elemental composition of XPS.
- For all discussion, please include a reference to support the authors’ claim.
- Please relate the crosslinking degree with the membrane thickness and performance.
Author Response
Response to the comments by Reviewer #1
- Comment: In introduction, please expound why CTA was chosen as support membrane to investigate its membrane formation.
Response and revision: CTA is a popular membrane material derived from the natural cellulose. Thus, CTA membranes attract extensive attention due to its great biocompatibility, excellent hydrophilicity, and outstanding antifouling property as compared with most of other polymeric membranes. (Page 4, line 1 from the bottom; page 5, lines 1-3)
- Comment: From Figure 1, membrane prepared by NIPS looks rougher than membrane prepared by TIPS. Please recheck the AFM analysis.
Response: We appreciate the reviewer’s comment. We further checked the membrane surface morphology. The membrane prepared by TIPS is rougher than that prepared by NIPS. This result is in consist with our previous work (Journal of Applied Polymer Science 2017, 134, 44454; Polymer 2018, 153, 150-160; Cellulose 2019, 26, 3747-3762).
- Comment: How did the author measure the membrane pore size? wet membrane has different pore size with dry membrane.
Response and revision: The surface pore size was determined by analyzing the FESEM images. (Page 7, line 7) As a result, the surface pore size should be corresponded to the dry membrane, and may be smaller than that of wet membrane.
- Comment: In fact, high surface porosity membrane should have higher surface roughness. This data was not fit with AFM. Please check.
Response: The surface roughness is not only affected by the surface porosity, but also influenced by the phase separation behavior. The CTA-TIPS membrane was formed by the solid-solid phase separation process, in which both CTA and DMSO2 crystalized under cooling. Thus, the surface of CTA-TIPS membrane was mainly constructed by the CTA crystalline region. In contrast, the CTA-NIPS membrane was formed by the mass exchange of water and DMSO2/PEG, which induced the solidification of CTA and resulted in a relatively smooth surface. On the other hand, both CTA-TIPS and CTA-NIPS membranes showed a small surface porosity (< 6%), whose effect on the surface roughness should be weak.
- Comment: Figure 4b, please translate to table for, and calculate the crosslinking degree. This work used MPD and TMC as monomer, the crosslinking degree can be easier to calculate from elemental composition of XPS.
Response and revision: The surface element content of TFC-FO membranes was further calculated by XPS result (Table 2). According to the O/N ratio, the cross-linking degree of PA on TFCNIPS is higher than those of TFCTIPS and TFCN-TIPS. (Page 14, lines 1-3 from the bottom)
Table 2 Surface element content of TFC-FO membranes measured by XPS
Samples |
C(%) |
N(%) |
O(%) |
S(%) |
O/N |
TFCTIPS |
71.00 |
9.78 |
19.03 |
0.19 |
1.94 |
TFCNIPS |
71.21 |
11.54 |
17.10 |
0.15 |
1.48 |
TFCN-TIPS |
70.12 |
10.24 |
19.37 |
0.27 |
1.89 |
- Comment: For all discussion, please include a reference to support the authors’ claim.
Response and revision: Several new references were cited in Results and Discussion section. “The performance of TFC-FO membranes can be connected to the morphology and the surface composition of PA layer, including the surface structure, the thickness and the cross-linking degree.36, 37” (Page 18, lines 1-3)
“In contrast to the dense PA layer of TFCTIPS membrane, the nodular protuberance and voids facilitate to increase the water transporting area and then enhance the water flux of TFC-FO membranes.34, 35” (Page 15, lines 1-3 from the bottom)
“The adsorbed MPD within the porous substrates will further diffuse out, react with the residual acyl chlorides, and form a thick PA film.33” (Page 15, lines 6-8)
- Comment: Please relate the crosslinking degree with the membrane thickness and performance.
Response and revision: The performance of TFC-FO membranes can be connected to the morphology and the surface composition of PA layer, including the surface structure, the thickness and the cross-linking degree [New Journal of Chemistry 2018, 42 (16), 13382-13392, Journal of Membrane Science 2016, 511, 29-39]. Compared with TFCTIPS membrane, the higher water flux for TFCN-TIPS and TFCNIPS membranes can be contributed to their intensive water transporting area from the nodular protuberance and voids inside the PA layer although their PA layers are thicker (Fig. 5). On the other hand, the severe reverse salt flux of TFCTIPS membrane may be ascribed to the low cross-linking degree of PA layer. (Page 18, lines 1-7)

Reviewer 2 Report
This is an interesting study focused on the influence of the porous support of PA TFC FO membranes on active layer structure obtained by interfacial polymerization and resulting separation performance. Cellulose triacetate (CTA) is chosen as membrane polymer for the support and three modes of casting cum phase separation (PS) were studied. The results support the hypothesis that a combined PS process, thermally and (liquid) nonsolvent induced, might lead to the best compromise between attractive features of the two other PS versions. However, the reasons for best overall performance are more complex; anyway, this is relatively well studied and discussed. Overall, I find the work rather well performed with view on the main goals, but rather “pragmatic” regarding the choice of PS conditions and limited in terms of scientific understanding of how and why the different CTA supports are obtained. That may be ok, if the deficits and errors are fixed and of the authors clearly state what they know and what they do not know. Overall, they may clearly express that the study is just an empirical one regarding the CTA membrane formation and that the study is scientifically focused on the resulting PA TFC FO membranes.
The key weakness of the paper is that it remains fully unclear why the specific PS conditions presented in Table 2 have been chosen; I did not find relevant references and the authors did not report own experiments.
The most important question: How did they know that a 10% CTA solution in “DMSO/PEG400 mixture” will undergo phase separation upon cooling from 160°C to 50°C?
Another question: What is the mechanism of the thermally induced phase separation? They write about “crystallization of CTA” as mechanism (see p. 6), but this is sheer speculation without any experimental proof (see also below).
Specific comments:
p. 2, 1st §: “As a consequence, water permeability” must be changed to “As a consequence, water flux”.
p. 2, 2nd§: The overview in Table 1 is too simplistic. For example, asymmetric membranes can also have sufficient stability because it is possible to prepare such membranes via NIPS also without macrovoids. Or, the “tendency of defect formation” is largely irrelevant if the membrane is only used as support for a TFC membrane. I recommend to omit the Table.
p. 3: sub-section numbering is wrong; should read “2.2.1” and “2.2.2”.
p. 6: In order to prove the evoked crystallization of CTA, quantitative XRD data for all three membranes must be reported and discussed.
p. 6: The statement that quenching from 160°C into water at 50°C will lead to PS by a combined mechanism (NIPS and TIPS) is sheer speculation without any scientific argument provided by the authors.
p. 8, 2nd §: It must read “amide I” and “amide II”!
p. 8, 3rd§: It must read “absorbed water”
p. 10, 1st §: The correct terminology for the two different orientations (“AL-DS” and AL-FS” had been introduced in Experimental on p. 4); this must also be used here, instead of the mis-leading terminology “FO mode” vs. “PRO mode” must be removed (in both orientations, FO experiments are performed!).
p. 10, 1st § and Table 3: Numbers should only be reported with significant digits, considering the experimental error.
p. 11, 2nd §: “via non-solvent induced phase separation and non-solvent induced phase separation” should be changed to “via non-solvent induced phase separation and thermally induced phase separation”.
Author Response
Dear reviewer,
Thank you very much to review our manuscript entitled "Preparation and properties of thin film composite forward osmosis membranes supported by cellulose triacetate porous substrate via nonsolvent-thermally induced phase separation process" (membranes-1640638). Following the comments, we revised the manuscript carefully, and the changes we made were marked in the revised version and listed as follows.
- Comment: The key weakness of the paper is that it remains fully unclear why the specific PS conditions presented in Table 2 have been chosen; I did not find relevant references and the authors did not report own experiments.
Response and revision: We appreciate the reviewer’s kind comment. The phase separation conditions presented in Table 2 were chosen according to our previous experiments. We investigated the formation of the CTA porous membranes by thermally induced phase separation (Journal of Applied Polymer Science 2017, 134, 44454, 1-10; Polymer 2018, 153, 150-160) or non-solvent thermally induced phase separation (Cellulose 2019, 26, 3747-3762) in our previous work.
“Then, three kinds of phase separation methods were introduced to obtain different CTA porous substrates, and the corresponding preparation conditions were list in Table 1 according to our previous work.22, 27, 28 (Page 6, lines 7-9)
- Comment:The most important question: How did they know that a 10% CTA solution in “DMSO2/PEG400 mixture” will undergo phase separation upon cooling from 160°C to 50°C?
Response: The thermally induced phase separation behavior of CTA/DMSO2/PEG400 system has been studied in our previous work (Journal of Applied Polymer Science 2017, 134, 44454). The phase separation diagram of CTA/DMSO2/PEG400 system was shown as follows. The solid lines divide the phase diagram into three regions: liquid region, liquid-solid region, and solid region. Theoretically, the CTA/DMSO2/PEG400 system will undergo liquid-solid phase separation as the temperature is lower than 80 °C. On the other hand, the formation of porous structure in CTA membrane further verified the happening of the phase separation of CTA/DMSO2/PEG400 mixture when cooled from 160 °C to 50 °C.
Figure R1. Phase diagram of CTA/DMSO2/PEG400 system. The solid squares (■), solid triangles (▲), and hollow circles (○) in (c) represent the melting points and crystallization temperatures measured by DSC, respectively. The solid lines divide the phase diagram into three regions: liquid region, liquid-solid region, and solid region. The dash line represents the dynamic solidification temperature upon cooling at a rate of 10 °C/min. (Journal of Applied Polymer Science 2017, 134, 44454)
- Comment:What is the mechanism of the thermally induced phase separation? They write about “crystallization of CTA” as mechanism (see p. 6), but this is sheer speculation without any experimental proof (see also below)
Response: The thermally induced phase separation behavior of CTA/DMSO2/PEG400 system has been studied in our previous work (Journal of Applied Polymer Science 2017, 134, 44454). The phase separation diagram of CTA/DMSO2/PEG400 system was shown in Figure R1. The solid lines divide the phase diagram into three regions: liquid region, liquid-solid region, and solid region. The CTA/DMSO2/PEG400 ternary system thermodynamically undergoes a liquid–solid phase separation. However, as a result of dynamics, the ternary system can be easily overcome the narrow liquid-solid region at a large cooling rate, and goes through a solid-solid phase separation.
- Comment: p. 2, 1st §: “As a consequence, water permeability” must be changed to “As a consequence, water flux”
Response and reversion: The sentence of “As a consequence, water permeability” was changed to “As a consequence, water flux”. (Page 3, lines 4-5 from the bottom)
- Comment:p. 2, 2nd§: The overview in Table 1 is too simplistic. For example, asymmetric membranes can also have sufficient stability because it is possible to prepare such membranes via NIPS also without macrovoids. Or, the “tendency of defect formation” is largely irrelevant if the membrane is only used as support for a TFC membrane. I recommend to omit the Table.
Response and revision: Table 1 has been omitted.
- Comment:p. 3: sub-section numbering is wrong; should read “2.2.1” and “2.2.2”.
Response and revision: The sub-section numbering was changed to “2.2.1” and “2.2.2”. (Page 6)
- Comment:p. 6: In order to prove the evoked crystallization of CTA, quantitative XRD data for all three membranes must be reported and discussed.
Response: We appreciate the reviewer’s comment. The crystallization of CTA has been carefully studied by XRD in our previous work (Journal of Applied Polymer Science 2017, 134, 44454). As seen in Figure R2, the WAXD patterns for the typical samples of CTA membranes present characteristic diffraction peaks around 17.0° and 23.0°, corresponding to (021) and (030) phase, respectively. They are as same as those shown in the pattern of CTA powder, indicating the crystallization of CTA during the phase separation.
Figure R2. WAXD patterns of CTA powder and CTA membranes.
- Comment:p. 6: The statement that quenching from 160°C into water at 50°C will lead to PS by a combined mechanism (NIPS and TIPS) is sheer speculation without any scientific argument provided by the authors.
Response: The nonsolvent-thermally induced phase separation behavior of CTA/DMSO2/PEG400 system has been studied in our previous work (Cellulose 2019, 26, 3747-3762). As seen in Figure R3, both NIPS and TIPS can be induced as the mixture was quenching from 160 °C into water at 50 °C.
Figure R3. The relationship between the morphology of CTA porous membranes and
the coagulation bath temperature and polymer concentration.
- Comment: p. 8, 2nd §: It must read “amide I” and “amide II”
Response and revision: The words “amine I” and “amine II” were changed to “amide I” and “amide II”. (Page 14, lines 9-10)
- Comment: p. 8, 3rd§: It must read “absorbed water”
Response and revision: The sentence of “w0 and w1 are the weights of water saturated and dried substrate (g), respectively” was changed to “w0 and w1 are the weights of substrate (g) before and after absorbing water, respectively”. (Page 8, lines 8-9)
- Comment: p. 10, 1st §: The correct terminology for the two different orientations (“AL-DS” and AL-FS” had been introduced in Experimental on p. 4); this must also be used here, instead of the mis-leading terminology “FO mode” vs. “PRO mode” must be removed (in both orientations, FO experiments are performed!).
Response and revision: The words “FO mode” and “PRO mode” were changed to “AL-FS” and “AL-DS”, respectively. (Page 9, line 2 from the bottom and page 17, lines 3-10 from the bottom)
- Comment: p. 10, 1st § and Table 3: Numbers should only be reported with significant digits, considering the experimental error.
Response and revision: The numbers “4.66667 ± 0.50083” and “10.02667 ± 0.57501” were changed to “4.67” and “10.03”, respectively. (Page 17, line 3 from the bottom) Similarly, some numbers in Table 3 were recalculated and reported with significant digits. (Page 19, Table 3)
- Comment: p. 11, 2nd §: “via non-solvent induced phase separation and non-solvent induced phase separation” should be changed to “via non-solvent induced phase separation and thermally induced phase separation”.
Response and revision: The word “nonsolvent” was changed to “thermally”. (Page 19, line 1 from the bottom and page 20, line 1)

Round 2
Reviewer 1 Report
The authors addressed well all of my concerns.
Reviewer 2 Report
The authors have well adressed the comments, the paper is now publishable.